# *Caenorhabditis elegans* as a Model System to Study Human Neurodegenerative Disorders

**DOI:** 10.3390/biom13030478

**Published:** 2023-03-05

**Authors:** Antonis Roussos, Katerina Kitopoulou, Fivos Borbolis, Konstantinos Palikaras

**Affiliations:** 1Department of Physiology, Medical School, National and Kapodistrian University of Athens, GR-157 27 Athens, Greece; 2Department of Biology, University of Padova, 35100 Padova, Italy

**Keywords:** ageing, Alzheimer’s disease, amyotrophic lateral sclerosis, Autosomal Dominant Optic Atrophy, *C. elegans*, Cockayne syndrome, Huntington’s disease, mitochondria, neurodegeneration, Parkinson’s disease

## Abstract

In recent years, advances in science and technology have improved our quality of life, enabling us to tackle diseases and increase human life expectancy. However, longevity is accompanied by an accretion in the frequency of age-related neurodegenerative diseases, creating a growing burden, with pervasive social impact for human societies. The cost of managing such chronic disorders and the lack of effective treatments highlight the need to decipher their molecular and genetic underpinnings, in order to discover new therapeutic targets. In this effort, the nematode *Caenorhabditis elegans* serves as a powerful tool to recapitulate several disease-related phenotypes and provides a highly malleable genetic model that allows the implementation of multidisciplinary approaches, in addition to large-scale genetic and pharmacological screens. Its anatomical transparency allows the use of co-expressed fluorescent proteins to track the progress of neurodegeneration. Moreover, the functional conservation of neuronal processes, along with the high homology between nematode and human genomes, render *C. elegans* extremely suitable for the study of human neurodegenerative disorders. This review describes nematode models used to study neurodegeneration and underscores their contribution in the effort to dissect the molecular basis of human diseases and identify novel gene targets with therapeutic potential.

## 1. Introduction

Battling human neurodegenerative diseases and their pervasive societal impact is becoming a global priority. Despite their devastating effect on human quality of life and their immense economic impact on healthcare systems, many human neurodegenerative conditions remain incurable and non-preventable. Ageing is universally associated with a marked decrease in neuronal function and higher susceptibility to neurodegeneration. In human populations, this is manifested as an ever-increasing prevalence of devastating neurodegenerative pathologies, such as Alzheimer’s disease (AD), Parkinson’s disease (PD), several ataxias and amyotrophic lateral sclerosis (ALS). Moreover, the steady increase in human lifespan in industrialized nations is exacerbating the prevalence of neurodegeneration-associated disability. Remarkably, neurodegenerative diseases are characterized by common hallmarks, including genomic instability, telomere attrition, epigenetic alterations, protein aggregate accumulation, mitochondrial dysfunction, stem cell exhaustion and autophagy defects, among others [1,2,3].

*Caenorhabditis elegans* is a soil dwelling, non-parasitic, free-living nematode that feeds on bacteria. The animals are of small size, with adults reaching approximately 1 mm in length and 80 μm in diameter, and can be easily cultivated in laboratory conditions, either on agar plates or in liquid medium, supplemented with *Escherichia coli* (mainly OP50 strain). *C. elegans* is primarily a hermaphroditic species, a feature that enables the generation of genetically identical populations and strain preservation, but males also exist with a frequency of <0.2%, allowing the implementation of genetic crosses. Each single hermaphrodite consists of 959 somatic cells and can produce around 300 progenies by self-fertilization, or 1000 upon mating with males (1031 somatic cells) [4,5,6]. Remarkably, the developmental and anatomical features of the nematode’s somatic cells have been extensively studied, and its entire cell lineage has been mapped [4,7]. It is also noteworthy that this animal is transparent at all stages of its life cycle, offering the unique ability for easy visualization, observation and monitoring of physiological and cellular processes by utilizing various microscopy techniques [8]. Additionally, *C. elegans* has a very short reproductive life cycle, which is completed in 2.5 days at 25 °C and 3.4 days at 20 °C, and its lifespan is about 2–3 weeks [5,6]. Its genome has been completely sequenced and annotated (~97 Mb size) since 1998, and it displays a high degree of conservation with human genes [9]. Overall, these characteristics render this nematode a versatile model organism for studying diverse biological phenomena and human disorders.

*C. elegans* is used as an animal model in major research areas, including developmental biology, neurobiology, cell biology and ageing, among others. Notably, it is estimated that around 42% of human disease-associated genes have a nematode ortholog, rendering this worm a suitable model to investigate the molecular mechanisms and cellular processes that orchestrate disease development and progression [10,11,12]. The most common experimental approaches to model a pathological condition in *C. elegans* are (1) to alter the expression levels of a disease-related homolog gene, or (2) to over-express a human isoform in specific nematode tissues, thus resulting in mutant and/or transgenic animals that display phenotypes reminiscent of human pathologies and can be easily studied or used in genetic screens [8,11,12]. Consequently, *C. elegans* has been established as an important model organism for neuroscience research, enabling the combination of diverse approaches, such as in vivo fluorescent imaging, neuronal activity manipulation and systematic genome-wide genetic screen, towards a common goal [8]. The nematode provides a unique opportunity to study neuronal function, neuronal circuit formation and neurodegeneration, because the entire connectome of its 302 neurons has been reconstructed and well-defined [13,14]. Additionally, its neurons share many functional characteristics with those of higher eukaryotes, including neurotransmitters, receptors, and neuromodulators [15,16,17,18]. Moreover, despite its simple nervous system, *C. elegans* shows a wide variety of responses that range from a simple aversion to mechanical stimuli to relatively complicated behaviors, such as defecation, thermotolerance, associative learning and memory [19,20,21,22]. Notably, age-associated decline of neuronal function is evolutionarily conserved in organisms as diverse as the nematode and humans, signifying commonalities in the underlying molecular mechanisms. Overall, *C. elegans* displays several advantages towards understanding the molecular mechanisms that drive the development and progression of age-associated neurodegeneration (Figure 1).

In this review, we present some of the most eminent nematode models of neurodegenerative pathologies and underline their importance in the elucidation of pathophysiological mechanisms and the identification of novel therapeutic intervention strategies against them (Figure 1).

### 1.1. Alzheimer’s Disease

Alzheimer’s disease (AD) is the leading cause of dementia worldwide and represents a major public health concern [23]. Predictions mention that by 2030, the prevalence of AD will reach 66 million cases, and by 2050, it is indicated that the prevalence of dementia will be tripled worldwide [24]. Although AD is strongly connected with heritability, at a percentage of 60–80%, several lifestyle and environmental factors can contribute to disease development and progression [24,25,26]. AD pathology starts with mild cognitive impairments that gradually evolve into behavioral and neuropsychiatric changes and deficits. Eventually, the progressive cognitive decline and the deterioration of social skills affect the ability of patients to function independently.

At the cellular level, the presence of insoluble amyloid β-peptide (Aβ) plaques and tau-associated neurofibrillary tangles (NFTs) in the brain are the most well appreciated pathological features of AD [24,25,26]. Aβ is a small peptide that is generated through the proteolysis of amyloid precursor protein (APP) by β- and γ-secretases. Although recent evidence indicates that the Aβ cascade hypothesis is debatable [27], Aβ plaques play a major role in the progression of the disease [28], while mutations in the APP gene, along with those in presenilin 1 (PSEN1) and presenilin 2 (PSEN2) genes, are linked to early-onset familial forms of AD [29,30]. Tau is a protein that binds to microtubules and stabilizes them, thereby supporting cellular function and viability via vesicle and organelle transportation along neuronal compartments [31,32,33]. Despite its beneficial and pivotal role in neuronal physiology, hyperphosphorylation of tau triggers its aggregation and leads to the formation of NFTs, while both wild type (wt) and mutant isoforms of tau are found to be hyperphosphorylated in AD patients [34,35,36].

#### 1.1.1. Aβ Models

Many *C. elegans* strains have been created over the years to recapitulate human AD pathology [37,38]. Among these, a variety of transgenic AD models have been generated by expressing human Aβ peptide in specific cell types, such as body wall muscle cells and neurons [39]. Notably, nematodes expressing Aβ_1–42_ peptide in body wall muscle cells exhibit accumulation of Aβ toxic oligomers and age-dependent progressive paralysis [40]. Likewise, AD models that overexpress human Aβ_1–42_ specifically in glutamatergic neurons demonstrate progressive, age-dependent neurodegeneration and have been used to validate the functional link between Aβ toxicity and endocytic trafficking [41], while models with pan-neuronal Aβ_1–42_ expression that demonstrate neuromuscular and age-dependent behavioral defects have contributed to the understanding of AD metabolic pathogenesis by revealing that reduced ATP levels and deficits in electron transport chain (ETC) complexes precede global metabolic failure [42]. Moreover, growing evidence indicates that accumulation of damaged mitochondria is a hallmark of age-dependent neurodegeneration and plays a detrimental role in the pathogenesis of AD [43]. Recent studies indicated that mitophagy, a selective form of autophagy that targets damaged and/or superfluous mitochondria, is impaired in AD patients and relevant animal models, including *C. elegans.* Indeed, the administration of potent mitophagy inducers, such as nicotinamide adenine dinucleotide (NAD^+^) boosters or urolithin A (UA), reverses memory impairment and cognitive deficits in Aβ- and tau-expressing *C. elegans* models [36]. These findings provide new insights into the mechanisms of mitophagy and neurodegenerative diseases and establish *C. elegans* as a platform for screening mitophagy modulators with therapeutic potential in AD.

#### 1.1.2. APOE Models

Apolipoprotein E (APOE) is thought to be responsible for the transport of cholesterol to neurons via astrocytes [44,45], and its isoforms are characterized as risk factors for AD development [46]. Although APOE*e2* isoform is linked with reduced risk, APOE*e4* is connected with increased risk of disease development and has been suggested to exacerbate early- and late-onset forms of AD [47], whereas APOE*e3* is neutral [48]. The generation and analysis of transgenic *C. elegans* models expressing human APOE alleles, with or without the presence of Aβ_1–42_, has provided valuable insights into the involvement of these isoforms in AD pathogenesis [49]. A recent study showed that the co-expression of APOE*e2* and Aβ results in the protection of glutamatergic neurons from degeneration and restores mechanosensory behavior. However, APOE*e4* and Aβ co-expression does not protect against Aβ neurotoxicity, while the co-expression of APOE*e3* has an intermediate phenotype [11]. APOE alleles do not affect neuronal function in the absence of the Aβ, highlighting their neuroprotective function. Indeed, the neuroprotective effects of specific APOE alleles can be modulated by pharmacological and/or genetic manipulations of endoplasmic reticulum (ER)-associated Ca^2+^ [49]. Notably, the decreased lifespan of *C. elegans* strains expressing APOE*e4* and Aβ can be rescued by the simultaneous expression of APOE*e2* or APOE*e3* [49]. In conclusion, *C. elegans* recapitulates the degeneration phenotypes linked to APOE polymorphisms and, thereby, can be used as a model to uncover new directions in AD research.

#### 1.1.3. APP Models

Aβ is a small peptide produced by the sequential enzymatic processing of APP. The *C. elegans* genome encodes for a single APP protein (APL-1), which is an ortholog of human APLP1-2 (amyloid beta precursor-like) proteins [50]. Single-copy pan-neuronal expression of human APP, in addition to the endogenous ALP-1 of the nematode, results in neurodegeneration and neurobehavioral dysfunction [51]. Similarly, the pan-neuronal overexpression of endogenous *apl-1* gene promotes memory, neurobehavioral and sensory plasticity deficits [52]. Conversely, the knockdown of *apl-1* gene causes rapidly progressing paralysis [53]. Investigation of APL-1 in the adult nervous system may provide further insights into the molecular function of APP and the pathways in which it is involved [54]. In conclusion, using *C. elegans* as a model to study APP activity could lead to a better understanding of its role in disease development and progression.

#### 1.1.4. Presenilin Models

Presenilins (PSENs) are transmembrane proteins found predominantly in the ER and are enriched in compartments that are in contact with mitochondria [55,56]. PSENs are a component of the γ-secretase complex, which plays a key role in the cleavage of APP into Aβ peptides [57]. Although mutations in PSEN1 and PSEN2 lead to early-onset familial AD, their functional consequences are not yet well-defined [58]. In congruence with the mammalian studies, mutation of the PSEN ortholog *sel-12* in *C. elegans* results in the dysregulation of ER Ca^2+^ homeostasis [58,59]. Particularly, *sel-12* mutants have elevated ER–mitochondrial Ca^2+^ signaling, triggering increased mitochondrial Ca^2+^ content, which subsequently leads to enhanced mitochondrial superoxide production. Moreover, *sel-12* mutants show mitochondrial metabolic defects that promote neurodegeneration [60]. However, the reduction of ER Ca^2+^ release, mitochondrial Ca^2+^ uptake or mitochondrial superoxide in mutant worms prevents neurodegeneration and rescues mitochondrial metabolic defects [60]. These findings suggest that mutations in PSEN alter mitochondrial metabolic function via ER–mitochondrial Ca^2+^ signaling, providing insights for new targets in the effort to tackle neurodegenerative diseases.

#### 1.1.5. Tau Models

The tau protein binds and stabilizes microtubules in order to support the neuronal cytoskeleton network [61]. In AD, tau becomes abnormally phosphorylated, leading to its aggregation and the formation of NFTs [62]. These molecular events mediate the blockage of the neuronal transport system and, thereby, impair synaptic communication between neurons. In *C. elegans*, *ptl-1* encodes for a tau-like protein that has 50% homology to mammalian tau [63]. Loss of *ptl-1* triggers incompletely penetrant lethality during embryogenesis, decreases lifespan, impairs touch sensitivity, and causes abnormal morphology in ALM touch neurons, recapitulating some aspects of AD pathology [64,65]. Accumulating evidence indicates that there is some functional conservation between tau and PTL-1, as human tau can rescue touch insensitivity in ptl-1 mutants, while defects caused by tau expression are ameliorated in the absence of endogenous PTL-1 [65]. There are many *C. elegans* models that overexpress either wt or mutant forms of the human tau protein, with the latter exhibiting greater toxicity [66,67]. Pan-neuronal expression of tau in the nematode leads to the accumulation of insoluble phosphorylated aggregates, age-dependent neurodegeneration, locomotion defects and abnormal motor neuron morphology. Moreover, tau pathology and phenotypes seem to be dose-dependent, since worms with higher levels of pan-neuronal expression display more severe locomotion defects [68,69]. Notably, the pan-neuronal expression of *A152T* tau mutation, a rare risk factor for frontotemporal dementia (FTD) and AD, results in decreased lifespan, locomotion defects and excessive degeneration of GABAergic neurons in nematodes, mimicking several aspects of AD pathology [70].

Forward and reverse genetic screens in *C. elegans* have identified *sut-1* and *sut-2* as mediators of tau neurotoxicity. Indeed, *sut-2* overexpression intensifies tau-associated pathology, while *sut-2* knockdown protects against tau-induced neuronal dysfunction [71,72]. Likewise, a genome-wide RNAi screen [73] in a nematode model that overexpresses the human tau pan-neuronally, unveiled that the unfolded protein response of the ER (UPR^ER^) is a potential modulator of tau proteostasis [74]. Consequently, expressing XBP-1, the driving transcription factor of UPR^ER^, enhanced the clearance of tau aggregates and improved neuronal survival [74]. Congruently, animals expressing XBP-1 in their neurons or intestine seem to have protection against multiple proteotoxic species, including Aβ_1–42_ peptide. Further supporting this notion, neuronal expression of XBP-1s rescued the loss of chemotaxis defects in animals that expressed Aβ_1–42_ pan-neuronally [5,74]. Recent studies demonstrated that neuronal XBP-1s upregulates lysosomal genes and mediates the enhancement of lysosomal acidity and function in the intestine. Taken together, these findings suggest that intestinal lysosomal function is required for increased proteostasis and longevity in the XBP-1 over-expressing nematodes [75].

Emerging findings have shown that there is a strong association between tau pathology and mitochondrial dysfunction, but the chronological order of these phenomena remains elusive [76,77]. A very recent study utilized a nematode model expressing low levels of the wt human tau (PIR3 strain) and underlined that mitochondrial impairment represents an early pathological event in neuronal cells [78]. These animals demonstrate short lifespan, impaired neurotransmission, defective locomotion, abnormal accumulation of tau aggregates and increased neurodegeneration, recapitulating several pathological features of the human tauopathy [77]. Moreover, pan-neuronal tau-expressing nematodes displayed mitochondrial impairment and locomotion deficits during development and ageing [78]. These findings indicate the toxic effects of tau in organelle function and cellular physiology. Although PIR3 nematodes do not accumulate detectable levels of tau aggregates during larval stages, they show increased mitochondrial damage and locomotion defects [78]. Interestingly, chelation of Ca^2+^ through EGTA rescues pathological phenotypes by increasing mitochondrial membrane potential and improving the motility of the worms. These findings suggest that there is a positive correlation among mitochondrial function, Ca^2+^ homeostasis and neuronal survival, providing additional mechanistic insights into the primary cause of mitochondrial dysfunction in early stages of tauopathy [78].

In conclusion, *C. elegans* models used for tauopathies and AD can shed light on the underlying molecular mechanisms of tau pathology and be used as screening platforms to unravel novel genes and compounds leading to the development of novel therapeutic interventions.

## 2. Parkinson’s Disease

PD is the second most frequent neurodegenerative disorder, affecting around two out of 100 people over 65 years of age, and is most likely caused by interactions among genetic, epigenetic and environmental factors [79,80]. Notably, exposure to toxicants, such as pesticides (rotenone and paraquat), 1-methyl-4-phenylpyridinium (MPP+) and neurotoxins such as 6-hydroxydopamine (6-OHDA), has been linked to PD development [81,82]. In particular, rotenone, which acts as an inhibitor of mitochondrial complex I, has already been studied in nematodes [83,84]. Nematodes treated with rotenone displayed loss of dopaminergic neurons, growth and motility defects as well as mitochondrial deficits [83,84]. However, a novel synthesized compound, mitochonic acid 5 (MA-5) derived from plant hormone Indole-3-Acetic Acid (IAA), has been shown to significantly reduce mitochondrial ROS elevation and degeneration of dopaminergic neurons, probably through its interaction with mitofilin as previously shown in mammalian cells [81,82,83,84,85,86,87,88,89,90,91]. Symptoms of PD include motor and non-motor symptoms, such as rest tremor, bradykinesia, rigidity, loss of postural reflexes and depression, and can be mostly traced to the loss of dopaminergic neurons in the substantia nigra, which leads to a reduction in dopamine release [79,80]. At the cellular level, PD is characterized by the presence of Lewy bodies containing mostly alpha-synuclein (α-syn) [92]. However, many of the molecular pathways underlying PD pathology remain elusive.

### 2.1. α-Synuclein Models

The α-syn protein is normally found at presynaptic terminals and the nucleus of neurons, mostly abundant in the brain, and encoded by the *PARK1/SNCA* locus in humans [93]. Although *C. elegans* has orthologs of many PARK genes, these do not include PARK1/SNCA [81]. Thus, nematode α-syn models are based on the neuronal or non-neuronal expression of wt or disease-associated forms of the human α-syn. Interestingly, pathological phenotypes of such models are mostly influenced by the expression pattern of α-syn, rather than the type of protein (wt or disease-associated) [94]. Indeed, overexpression of α-syn pan-neuronally or in motor neurons causes age-independent motor defects. In contrast, these defects are not observed when α-syn is expressed only in dopaminergic neurons [94]. However, neuron loss is induced by pan-neuronal or dopaminergic neuron-specific α-syn expression, but not by its expression in motor neurons [94]. Moreover, such heterologous expression of α-syn has been shown to result in the formation of aggregates that resemble inclusions of human PD neurons, thus providing versatile platforms for the discovery of aggregation modulators. For instance, accumulation of both wt and mutant α-syn has been observed in the cell bodies and neurites of dopaminergic neurons in *C. elegans* [95].

Similarly, age-dependent formation of aggregated α-syn inclusions has been reported in worms expressing human α-syn tagged with yellow fluorescent protein (YFP) in body wall muscle cells [96]. An RNAi screen on this model uncovered 80 genes that cause premature accumulation of inclusions, when knocked down. A significant number of these genes are related to ageing, while 49 out of 80 identified genes have a human ortholog [96]. In this context, it was found that loss of *tdo-2* in *C. elegans*, coding for the tryptophan 2,3-dioxygenase (TDO-2), leads to lifespan extension, decreased α-syn toxicity and augmented tryptophan levels, indicating tryptophan-dependent toxicity regulation by TDO-2 [97]. Likewise, an RNAi screen in worms expressing α-syn in dopaminergic neurons unveiled five genes with potential neuroprotective roles. Most of them are genes correlated to vesicle trafficking [98]. These results, combined with the neuroprotective effect of TOR-2 (the nematode ortholog of human torsin family 1 member B) and mammalian Rab1A, a GTPase involved in ER-to-Golgi transport, indicate a correlation between impaired ER-to-Golgi vesicular transport and α-syn toxicity [99,100]. Moreover, the same model was used to display that heat preconditioning, mediated by the heat shock transcription factor HSF-1 and the small heat-shock protein HSP-16.1, decreases the loss of dopaminergic neurons [101]. HSP-16.1 interacts with PMR-1 (plasma membrane-related Ca^2+^—ATPase 1), and thereby, maintains Ca^2+^ homeostasis and prevents neuronal cell death [101]. Similarly, loss of CPS-6 (EndoG homologue in *C. elegans*) or depletion of the oleic acid (OA)-generating enzyme stearoyl-CoA-desaturase (SCD) in these animals also leads to decreased dopaminergic neurodegeneration, pointing to novel targets for PD treatment [102,103]. Notably, the mechanisms mentioned above are evolutionarily conserved, as similar results have been obtained from yeast, rodent and human neuronal models [103]. Finally, recent studies showed that intestinal over-expression of α-syn mutants leads to mitochondrial accumulation, excessive fragmentation and energetic stress, as well as increased activation of the mitochondrial unfolded protein response (UPR^mt^), thus establishing a link between the mitochondrial quality control system and α-syn toxicity [104].

### 2.2. LRRK2 Models

Leucine rich repeat kinase 2 (LRRK2) is a protein found mainly in the cytoplasm but also interacts with the outer mitochondrial membrane [105]. Mutations in the LRRK2 gene are the most common cause of PD, responsible for 4% of autosomal dominant cases [106]. Although the function of LRRK2 remains unknown, it has been associated with autophagy, cytoskeletal activity and vesicular trafficking [106]. The ortholog of LRRK2 in *C. elegans* (LRK-1) is localized in the Golgi apparatus and is expressed in muscle cells, neurons and intestinal cells [107,108]. LRK-1 deficient nematodes demonstrate abnormal distribution of synaptic vesicle proteins in neurons and display dopamine-specific behavioral defects [109]. Moreover, these worms exhibit impairments in the trafficking of the synaptic vesicles [110] and suppress phenotypes of PTEN-induced kinase (*pink-1*) loss-of-function worms [111]. There are also several transgenic nematode models that express normal or mutant forms of the human LRRK2 protein. Animals with pan-neuronal LRRK2 expression have been used to demonstrate that mutant forms of the protein induce greater loss of dopaminergic neurons compared to their wt counterparts [112]. Interestingly, dopaminergic neuronal loss is rescued by kinase inhibitors, suggesting that LRRK2 inhibition could be a possible treatment for PD [112,113]. Congruently, dopaminergic-specific overexpression of LRRK2 human variants has been shown to trigger age-dependent neuronal death, behavioral and locomotion defects, as well as a decrease in dopamine levels [107]. In conclusion, LRRK2 *C. elegans* models recapitulate several pathological features of PD, highlighting them as ideal platforms to investigate the molecular mechanisms that regulate LRRK2-mediated pathology.

## 3. Amyotrophic Lateral Sclerosis (ALS)

ALS is a fatal neurodegenerative disease and the most common motor neuron disorder; it shows an incidence between 0.6 and 3.8 per 100,000 persons every year and a prevalence between 4.1 and 8.4 per 100,000 individuals [114]. ALS is characterized by the degeneration of motor neurons in the spinal cord, motor cortex, corticospinal tracts and brainstem, leading to progressive muscular paralysis [115]. The majority of ALS cases are sporadic, with unknown etiology, while only about 10% are familial [116]. *C. elegans* has been used to investigate genes that play a crucial role in both sporadic and familial forms, including chromosome 9 open reading frame 72 (C9ORF72), superoxide dismutase 1 (SOD1), TAR DNA-binding protein (TDP43) and fused in sarcoma (FUS).

### 3.1. C9ORF72 Models

Expansions of GGGGCC hexanucleotide repeat in the C9ORF72 gene are the most prevalent genetic cause of ALS in Europe and North America [117,118]. Although the mechanism of this expansion is still unknown, a possible explanation is that AUG independent translation of GGGGCC leads to the formation of the respective dipeptide repeats [119]. Indeed, a *C. elegans* model expressing arginine-containing dipeptides exhibits age-dependent toxicity in muscle cells and motor neurons [120]. Interestingly, the pathogenetic threshold of the repeats that is required for the disease development and progression has not been identified. In most cases, ALS patients have hundreds of repeats [117,121]. Worms with loss-of-function mutations in the C9ORF72 ortholog *alfa-1* gene demonstrate age-dependent motility impairments, which eventually result in paralysis and GABAergic stress-dependent neurodegeneration [122]. Moreover, *alfa-1* mutants exhibit endocytosis defects that can be partially rescued by the expression of the human wt C9ORF72 protein, revealing a degree of functional conservation [123]. However, since *alfa*-*1* does not carry hexanucleotide repeat expansions, transgenic nematode models are mainly used to study human C9ORF72 toxicity. The use of such models has revealed that the transgenes containing 29 GGGGCC repeats result in early-onset paralysis and lethality, in contrast to those with 9 repeats, which had a less severe impact [124]. Moreover, a forward genetic screening in such transgenic animals revealed two genes that suppress C9ORF72-induced toxicity [124].

### 3.2. SOD1 Models

The SOD1 enzyme catalyzes the detoxification of superoxide. Although SOD1 mutations account for about 2% of familial ALS cases [125], the underlying toxicity mechanism remains to be uncovered [126]. *C. elegans* has five genes (*sod-1*, *sod-2*, *sod-3*, *sod-4* and *sod-5*) coding for superoxide dismutases [127]. Nevertheless, SOD1 toxicity has been mainly studied in humanized transgenic worms. In fact, worms expressing normal human SOD1 in neurons and/or in muscles demonstrate a normal phenotype [128,129]. By contrast, muscle-specific or pan-neuronal expression of SOD1 variants in *C. elegans* leads to ALS-like phenotypes. More specifically, these worms demonstrate increased protein aggregation, impaired locomotion, susceptibility to oxidative stress, synaptic deficiencies, axonal guidance defects and age-dependent paralysis [128,129,130]. The study of various single-copy SOD1 knock-in worm models, which reproduce mutations of ALS patients, revealed that both loss and gain of SOD1 function can promote the pathogenesis of ALS in distinct neurons [131]. Notably, loss-of-function mutations affect glutaminergic neurons, while SOD1 gain of function has a selective impact on cholinergic neurons [131]. Moreover, emerging findings indicate that metformin can protect nematodes expressing mutant SOD1, by activating autophagy, and cause a DAF-16-mediated lifespan extension [132].

### 3.3. TDP43 Models

TDP43 is a DNA/RNA-binding protein that regulates transcription, alternative splicing and subsequently gene expression [133]. In ALS-affected neurons, TDP43 is hyper-phosphorylated, ubiquitinated, truncated and localized in cytoplasmic inclusion bodies [134,135]. The ortholog of TDP43 in *C. elegans* is TDP-1, which is expressed in neurons, body wall muscle cells and the pharynx [136]. TDP-1 depleted nematodes demonstrate locomotion, fertility and growth defects, as well as increased sensitivity to oxidative and osmotic stress [136,137]. On the other hand, TDP-1 deficiency ameliorates proteotoxicity in various *C. elegans* models, including TDP43 and SOD1 expressing animals, and extends the lifespan of the nematode in a DAF-16/FOXO-dependent manner [136,137]. Moreover, the genetic ablation of TDP-1 rescues neuronal degeneration in a *C. elegans* model that overexpresses mutant TDP43 in GABAergic neurons [138]. Conversely, overexpression of TDP-1 by its native promoter reduces nematode lifespan, while neuronal overexpression of either endogenous TDP-1 or mutant human TDP43 causes uncoordinated locomotion, abnormal synaptic morphology, degeneration of GABAergic motor neurons and age-dependent motor dysfunction [138,139,140], indicating a dose-dependent effect of TDP43 on neuronal function and survival.

### 3.4. FUS Models

FUS encodes for an RNA/DNA binding protein and is found to be localized in the nuclear compartment and involved in the regulation of various cellular processes [141,142]. On the other hand, ALS-associated mutant FUS is accumulated in the cytoplasm and forms neurotoxic ribonucleoprotein granules and inclusions [143,144]. At the molecular level, mutant FUS triggers dysregulation of RNA processes, such as splicing, transcription and stabilization, leading subsequently to neuronal dysfunction [145,146,147].

The homolog gene of human FUS in *C. elegans* is *fust-1* and has been shown to be involved the regulation of neuronal integrity, synaptic function, lifespan and stress responses [148]. Even so, transgenic *C. elegans* models expressing human FUS are preferentially used in order to recapitulate ALS phenotypes. As a result, many nematode strains expressing either full length of the wt FUS protein or ALS-associated FUS variants have been generated. Such variants include both missense mutations (e.g., R514G, R521G, R522G, R524S, P525L) and truncations (e.g., FUS513 and FUS501), associated with the varying clinical severity of ALS patients [149]. Although worms expressing wt human FUS pan-neuronally are normal, expression of mutant forms results in the formation of cytoplasmic inclusions, age-dependent motor dysfunction and reduced lifespan [149]. Moreover, *C. elegans* models expressing different FUS variants display a range of phenotypic severity as seen in humans [150]. Indeed, nematodes expressing mutant FUS (FUSS57Δ) in GABAergic neurons exhibit neuronal dysfunctions, motor deficits and neurodegeneration, symptoms that are reminiscent of human ALS phenotypes [151]. Likewise, transgenic worms expressing an aggregation-prone FUS variant in GABAergic neurons display neurodegeneration, synaptic impairment and paralysis [138], while overexpression of an ALS-associated FUS mutation (FUS501) was recently reported to disrupt neuromuscular junction (NMJ) morphology, resulting in defective neuromuscular transmission and synaptic deformation [152]. Taken together, these findings indicate that *C. elegans* is an excellent model for biological and medical research to delineate the molecular mechanisms of ALS pathogenesis.

## 4. Huntington’s Disease

Huntington’s disease (HD) is an autosomal dominant neurodegenerative disorder characterized by motor, cognitive and psychiatric problems. HD is caused by a CAG expansion in the huntingtin (HTT) gene, resulting in the production of a mutant HTT protein that carries extended polyglutamine repeats (polyQs) in its N-terminus, triggering neuronal death [153,154]. Although there is no HTT ortholog in *C. elegans*, several HD models have been generated, carrying variable lengths of polyQ-repeats with or without fusion with fluorescent proteins. These models have been widely used to monitor the accumulation of polyQ tracts upon challenging conditions and during ageing in vivo. Both worms and humans require at least 35 to 40 polyQ repeats for the onset of aggregation, and it has been shown that in both cases the length of polyQ tracts is correlated with the age of onset and the severity of pathological features [155,156]. Particularly, transgenic nematodes expressing polyQ2–Q150 in ASH neurons showed that long polyQ tracts (Htn-Q150) trigger age-dependent protein aggregation and gradual neuronal loss [157]. The expression of HTT N-terminus with 19, 88 or 128Qs in mechanosensory neurons further validated the length-dependent nature of polyQ toxicity, as Q128-expressing animals show axonal abnormalities and increased protein aggregation [158].

*C. elegans* HD models have been used to identify potential modulators of polyQ aggregation and toxicity. To this end, the cytoprotective effect against polyQ tracks of several naturally occurring and synthetic chemical compounds, such as Rutin and diphenyl diselenide (Ph_2_Se_2_), have been identified using the *C. elegans* HD models. Rutin supplementation promoted polyQ aggregate reduction, neuroprotection and lifespan extension through elevated autophagy, insulin/IGF1-like signaling (IIS) modulation [159] and increased antioxidant activity [160]. Further supporting the beneficial role of autophagy, Htn-Q150- or polyQ40-expressing nematodes displayed aberrant accumulation of the toxic polyQ aggregates, leading subsequently to increased neurodegeneration upon autophagy inhibition [161]. Chronic administration of Ph_2_Se_2_ resulted in diminished polyQ aggregation and subsequent neuroprotection via a molecular mechanism that is dependent on DAF-16, HSP-16.2 and SOD-3 to enhance antioxidant capacity and proteostasis [162]. Interestingly, glucose supplementation reduced the misfolded proteins and delayed neurodegeneration in polyQ128-expressing nematodes in a DAF-16 dependent manner [163]. During the last decade, nematode HD models have been used as screening platforms, resulting in the characterization of synthetic and/or natural bioactive agents to tackle polyQ cytotoxicity and neurodegeneration [155,156,157,158,159]. In addition to the neuronal polyQ-expressing nematodes, several *C. elegans* HD models have been generated to express polyQ peptides fused with fluorescent proteins in body wall muscle cells and, thereby, to assess the rate of polyQ aggregate formation. Such a newly developed HD model expressing polyQ128 tagged with YFP demonstrated increased cytotoxicity, motor deficits and reduced lifespan [164]. Recent studies showed that the polyQ expression in body wall muscle cells mediates increased mitochondrial network fragmentation, indicating an intricate association between polyQ toxicity and mitochondrial dynamics [165,166]. Interestingly, body wall muscle-expressing polyQ nematodes displayed improved locomotion upon knocking down several genes that reduced mitochondrial fragmentation [165,166]. In conclusion, *C. elegans* is a promising model organism for studying HD and other polyglutamine-based diseases [167].

## 5. Cockayne Syndrome

Cockayne syndrome (CS) is a rare autosomal recessive neurodegenerative disorder with an incidence of 2.5 cases per million [168]. The syndrome is characterized by premature ageing, dwarfism, mental retardation, microencephaly and severe photosensitivity. CS is caused by mutations in the ERCC6 and ERCC8 genes coding for Cockayne Syndrome group B protein (CSB) (accounting for 80% of cases) and Cockayne Syndrome group A protein (CSA) (accounting for approximately 20% of the cases), respectively. These genes play a vital role in the transcription coupled (TC) nucleotide excision DNA repair mechanism (NER) by initiating the cascade of events that occurs upon RNA polymerase II stalling and serves to remove helix distorting lesions, such as UV-induced cyclobutane pyrimidine dimers (CPDs) [169,170,171]. CSA and CSB proteins are evolutionarily conserved across species, and mutations in *csa-1* and *csb-1*, homologues of ERCC6 and ERCC8 human genes, respectively, have already been characterized in *C. elegans*, with CSA-1 and CSB-1 deficient nematodes displaying developmental growth retardation and lifespan shortening upon UV treatment [172,173,174]. Moreover, loss of *csb-1* causes reduced somatic tissue functionality and mechanosensory and neuronal defects along with progressive neurodegeneration upon UV-induced DNA damage [175]. Notably, in both human cells and *C. elegans*, CSB-1/CSB deficiency promotes accumulation of dysfunctional mitochondria and mitochondrial network hyperfusion, leading to altered energy metabolism [175]. Since *C. elegans* has many common disease manifestations with human pathology of CS, this model can shed light on the underlying role of DNA damage in age-associated progressive loss of neuronal integrity.

Conversely, CSA-1 seems dispensable for global genome nucleotide excision repair mechanisms (GG-NER), which act throughout the entire genome but target mostly helix distortions [171,176], since UV irradiation of *csa-1* mutant worms has no significant effect on germline maintenance, which requires active GG-NER [177]. However, CSA-1 participates in the regulation of TC-NER, which is vital in the transcriptionally hyperactive context of early larval development and cooperates with CSB-1 and XPC-1 to repair UV-induced DNA lesions [177]. As result, *csa-1* mutants are more sensitive to UV irradiation during larval stages and exhibit developmental defects and growth arrest, thus providing a powerful system to further explore and define the role of TC-NER during development and ageing [173]. Moreover, these observations indicate the requirement for TC-NER during development, and GG-NER in proliferating cell types is conserved in *C. elegans*, rendering nematode models suitable for the study of distinct developmental pathologies that are observed in CS patients and the investigation of the differential utilization of the two NER branches during development.

## 6. Autosomal Dominant Optic Atrophy (ADOA)

Autosomal Dominant Optic Atrophy (ADOA) is a rare genetic neurodegenerative disease that causes progressive and irreversible loss of vision in humans. At the molecular level, ADOA is caused by mutations in optic atrophy 1 (OPA1), an inner mitochondrial membrane protein that is implicated in the process of mitochondrial fusion [178,179,180]. Recent studies utilized *C. elegans* as an animal model for ADOA to understand the pathophysiological mechanisms of the disease. Mutations in the OPA1 homolog gene *eat-3*, or heterologous expression of mutant human OPA1^K301A^, lead to the accumulation of autophagosomes and mitochondrial dysfunction, ultimately causing decreased mitochondrial content in axons of GABAergic neurons, neuronal degeneration and impaired defecation cycle. Interestingly, autophagy inhibition restores axonal mitochondrial density and downstream pathological manifestations [178]. Moreover, overexpression of OPA1^K301A^ or EAT-3 deficiency triggered excessive and uncontrolled Ca^2+^-dependent mitophagy, leading to reduced mitochondrial content in axons, a response that is tightly dependent on Ca^2+^calcineurin-AMPK signaling cascade [181]. Congruently, Ca^2+^ chelation restores the defective autophagosomal and mitochondrial distribution in neuronal processes [181]. All of these findings, in combination with the promising results from rodents, underscore that inhibition of excessive autophagy/mitophagy and the modulation of AMPK activity could lead to the development of novel therapeutic interventions to improve or even rescue the visual deficits in ADOA patients. Interestingly, accumulating evidence suggests that mitochondrial dysfunction and mitophagy impairment are hallmarks of multiple neurodegenerative diseases [182], implying that such insights into the molecular mechanism of ADOA pathogenesis and the contribution of autophagy could also apply to the treatment of other neurodegenerative diseases.

## 7. Concluding Remarks

Despite the benefits of *C. elegans* for modeling human neurodegenerative diseases, some limitations of this model should be considered. The nematode has a simple nervous system, without myelin sheaths, and lacks many mammalian anatomical features, including a circulatory system and blood–brain barrier [183]. In addition, *C. elegans* does not have a first-pass liver metabolic pathway or a kidney that filters blood [183]. Although such limitations render *C. elegans* unable to completely summarize the pathophysiology of human neurodegenerative diseases, nematode models carry various benefits that are useful for this field of research (Figure 2). The ease of laboratory culture and manipulation, along with the short life cycle, make this model cost-effective and less time-consuming compared to others. Moreover, the application of unbiased forward and reverse genetic screening approaches, as well as its high susceptibility to RNAi and transgenesis, establish *C. elegans* as a malleable experimental tool. An additional advantage of the worm is found in its transparent body structure, which enables the tracking of fluorescent markers in vivo and the assessment of cellular and physiological processes [184]. Therefore, modeling a human disease in *C. elegans* allows the investigation of complex molecular pathways and the identification of their components. Last but not least, the nematode has no bioethical limitations, thus allowing the scientific community to implement fast-track experimental protocols and interventions that could not be applied to other animal models.

As a result, recent years have seen a significant increase in the use of *C. elegans* as a high-throughput screening platform for drug discovery [185,186,187]. This approach has led to the identification of several chemical compounds and/or small molecules with potential therapeutic benefits against neurodegenerative disorders [188]. The most renowned examples of such discoveries include the potential of using α-Methyl-α-phenylsuccinimide (MPS) as a treatment for TDP43-related proteinopathies, the identification of neuroleptics as promising anti-ALS compounds that stabilize neuromuscular transmission, and the suggestion that LRKK2 inhibitors can be used as effective drugs against PD [113,189,190]. Likewise, nematode models have been extensively used for the in vivo validation of anti-neurodegenerative properties of drugs, suggested by in vitro assays and/or screens in unicellular organisms. Such a workflow has been successfully applied to uncover the beneficial effect of clioquinol (CQ) on Aβ aggregation and toxicity, as well as in the discovery of SynuClean-D, CNS-11 and CNS-11g, small compounds that inhibit α-syn aggregation and mitigate its toxicity, among other instances [191,192,193]. Moreover, the development of microfluidics devices and the recent integration of machine learning algorithms have provided additional improvements to nematode-based drug discovery methodologies by facilitating the design, execution and analysis of more elaborate screens. Such advanced approaches have led to the identification of novel substances with therapeutic potential, such as the FDA-approved clinical compounds tofranil, dronedarone, bendrofluazide and buspar, which hold promise as anti-HD drugs, enasidenib, ethosuximide, metformin and nitisinone as candidates for late PD treatment, as well as the natural compounds Kaempferol and Rhapontigenin, which induce mitophagy and improve cognition in AD models [194,195,196].

Finally, the powerful genetics of *C. elegans* have contributed immensely to the unveiling of molecular mechanisms that underly the activity of candidate drugs, as well as to the discovery of key endogenous molecules and processes that are involved in disease pathogenesis and consist of potential targets for therapeutic interventions. Such discoveries include the function of UPR^ER^ as a modulator of tau aggregation, the participation of SUT-1 and SUT-2 in the activation of tau, the protective effect of the glycolytic enzyme GPI against *α*-syn toxicity, the tendency of dopamine to exacerbate α-syn induced neurotoxicity, as well as the nature of autophagy and mitophagy as substantial factors in the pathogenesis of PD and other neurodegenerative disorders [72,73,74,98,197,198,199,200,201]. Overall, the aforementioned advantages establish *C. elegans* as a powerful and versatile preclinical model, with a great contribution to the understanding of the pathophysiological mechanisms of age-associated neurodegeneration.

## Figures and Tables

**Figure 1 biomolecules-13-00478-f001:**
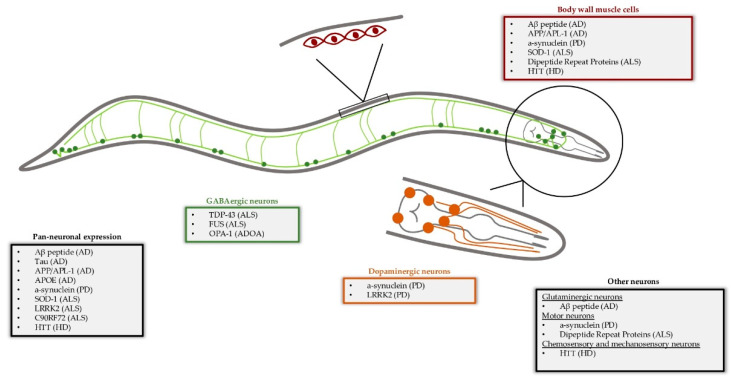
Examples of *C. elegans* models used to study neurodegenerative disorders by expressing human disease-associated genes in specific cell types. The GABAergic neurons are illustrated in green, and magnified are the body wall muscle cells (red) and the dopaminergic neurons (orange). AD = Alzheimer’s disease, PD = Parkinson’s disease, ALS = amyotrophic lateral sclerosis, HD = Huntington’s disease, ADOA = Autosomal Dominant Optic Atrophy.

**Figure 2 biomolecules-13-00478-f002:**
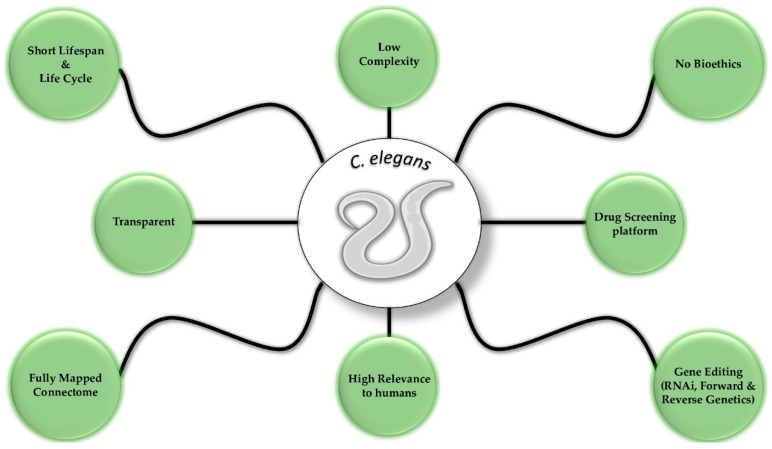
Selected advantages of *C. elegans* that render it an ideal animal model.

## Data Availability

No new data were created or analyzed in this study. Data sharing is not applicable to this article.

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
