# Peer review of "Caenorhabditis elegans as a Model System to Study Human Neurodegenerative Disorders"

_biomolecules, 2023, doi:10.3390/biom13030478_

Round 1

Reviewer 1 Report

The authors presented an excellent review of neurodegenerative models in C. elegans. The references are relevant, and the models are not frequently discussed in the published reviews. I really recommend the paper for publication.

•The authors can find below some suggestions and misspelling words to correct:

• The aim in both the abstract and introduction: I recommend the authors specify the specie described in the review. Other nematode species are used as models for neurodegenerative diseases, but the authors only discussed the C. elegans models. 

•Mispelling words:

•a-syn.

•In vivo: regular, not italic.

•remove the extra spaces: e.g., “through elevated autophagy, insulin/IGF1-like signaling (IIS).”

•Standardize the references.

Author Response

The authors presented an excellent review of neurodegenerative models in C. elegans. The references are relevant, and the models are not frequently discussed in the published reviews. I really recommend the paper for publication.

We thank the Referee for the encouraging comment.

The authors can find below some suggestions and misspelling words to correct:

The aim in both the abstract and introduction: I recommend the authors specify the specie described in the review. Other nematode species are used as models for neurodegenerative diseases, but the authors only discussed the C. elegans models. 

Our review focuses only on Caenorhabditis elegans nematode as it is already stated in our title. Per the suggestion of the Referee, we have now revised the text accordingly to make it clear.

Reviewer 2 Report

The authors have summarized in a review article the latest findings on C. elegans as a model system for studying human neurodegenerative diseases. The review is very significant, well organized, and will be of importance to a wide reader of Biomolecules.

For more detailed comments, please see the attachment.

Author Response

The authors have summarized in a review article the latest findings on C. elegans as a model system for studying human neurodegenerative diseases. The review is very significant, well organized, and will be of importance to a wide reader of Biomolecules.

We thank the Referee for the encouraging comment.

For more detailed comments, please see the attachment.

The authors have summarized in a review article the latest findings on C. elegans as a model system for studying human neurodegenerative diseases. The review is very significant, well organized, and will be of importance to a wide reader of Biomolecules.

As a major comment, the authors might want to add a disease model of PD by rotenone treatment in section 2.3? How about citing below the effect of low concentration rotenone treatment as a model of PD in animal cells and nematodes and the improvement of nematode PD models by MA-5, a novel therapeutic candidate for mitochondrial disease?

(Related references)

  • Human neuroblastoma cells “Rotenone Models”

doi: 10.1523/JNEUROSCI.23-34-10756.2003

  • Mitofilin overexpression attenuates vulnerability of dopaminergic cells to dopamine and rotenone.https://doi.org/10.1016/j.nbd.2016.03.015

Parkinson’s disease-related stressors (Dopamine and Rotenone) >>>>

Parkin protein is involved in the degradation of Mitofilin in response to DA stressor.

  • Am J Transl Res. 2020 Nov 15;12(11):7542-7564. eCollection 2020.
  1. elegans “Rotenone model”
  • Mocko JB. et. al, 2010 DOI: 10.1016/j.nbd.2010.03.019
  • Zhou S et al., 2013 Int J Biochem Mol Biol v.4(4); 2013 PMC3867705

(Recent result)

Mitochonic Acid-5 (MA-5) attenuates C. elegans Rotenone model.

  • Wu X et al., Int J Mol Sci. 2022 Aug 24;23(17):9572. doi: 10.3390/ijms23179572.

(Mitochonic acid-5 in mammalian cells)

  • MA-5 interacts with mitofilin and modulates the mitochondrial inner membrane organizing system (MINOS) in cultured mammalian cells. J Am Soc Nephrol. 2016 Jul;27(7):1925-32. doi: 1681/ASN.2015060623.
  • MA-5 has potential as a drug for the treatment of various mitochondrial diseases. 2017 Jun;20:27-38. doi: 10.1016/j.ebiom.2017.05.016.

MA-5 is a newly synthesized derivative of the plant hormone, IAA.

  • Tohoku J Exp Med. 2015 Jul;236(3):225-32. doi: 10.1620/tjem.236.225.

(as related review articles)2

  • Johnson and Bobrovskaya

https://doi.org/10.1016/j.neuro.2014.12.002

  • Cooper JF and Van Raamsdonk JM

doi: 10.3233/JPD-171258

Following the suggestion of the Referee, we have now included a paragraph describing the impact of rotenone in the survival of dopaminergic neurons. Moreover, we demonstrated the neuroprotective effect of mitochonic acid-5.

Typo as minor comment.

page 2, line 3 “----- approximately 1 mm in length and 0.8 μm in diameter,”

page 2, line 3 “----- approximately 1 mm in length and 80 μm in diameter,”

Corrected

Reviewer 3 Report

Roussons et al. described what have been done for disease modeling in C. elegans. They described 6 major neurodegenerative diseases in humans. They mention what are known for human diseases and then described C. elegans models focusing on the disease susceptible genes; loss of function and over-expression etc.

Generally, the conclusion of the manuscript is C. elegans is a very good model organism for disease study, although they mention the differences of C. elegans and humans such as body morphology. Unfortunately, the style of the manuscript is mostly limited to the catalogue of the recapitulation of human diseases in C. elegans. I strongly suggest that what are the new findings by using C. elegans as disease models should be stressed. For example, by using genetic screen with forward, reverse genetics and RNAi screening, new molecules that are important pathogenesis are found in C. elegans model for the first time, should be incorporated. Also, the disease model is not only pathogenesis study, but also to search for solution. Authors should explain how those aims have been achieved; for example, some compounds are proposed for the first time using such models. 

Author Response

Roussons et al. described what have been done for disease modeling in C. elegans. They described 6 major neurodegenerative diseases in humans. They mention what are known for human diseases and then described C. elegans models focusing on the disease susceptible genes; loss of function and over-expression etc. Generally, the conclusion of the manuscript is C. elegans is a very good model organism for disease study, although they mention the differences of C. elegans and humans such as body morphology.

Unfortunately, the style of the manuscript is mostly limited to the catalogue of the recapitulation of human diseases in C. elegans. I strongly suggest that what are the new findings by using C. elegans as disease models should be stressed. For example, by using genetic screen with forward, reverse genetics and RNAi screening, new molecules that are important pathogenesis are found in C. elegans model for the first time, should be incorporated.

We would like to thank the Referee for the valuable comments. We have now changed the text accordingly.

Also, the disease model is not only pathogenesis study, but also to search for solution. Authors should explain how those aims have been achieved; for example, some compounds are proposed for the first time using such models. 

In the revised manuscript we have now highlighted in the section of “Concluding remarks” the use of C. elegans as a versatile and high-throughput screening platform and the identification of several chemical compounds and/or small molecules with potential therapeutic benefits against neurodegenerative disorders with the respective references.

Round 2

Reviewer 3 Report

The authors addressed the concerns appropriately.